# Probing the Influence of Novel Organometallic Copper(II) Complexes on Spinach PSII Photochemistry Using OJIP Fluorescence Transient Measurements

**DOI:** 10.3390/biom13071058

**Published:** 2023-06-29

**Authors:** Sergei K. Zharmukhamedov, Mehriban S. Shabanova, Irada M. Huseynova, Mehmet Sayım Karacan, Nurcan Karacan, Hande Akar, Vladimir D. Kreslavski, Hesham F. Alharby, Barry D. Bruce, Suleyman I. Allakhverdiev

**Affiliations:** 1Institute of Basic Biological Problems, FRC PSCBR RAS, 142290 Pushchino, Russia; 2Bionanotechnology Laboratory, Institute of Molecular Biology and Biotechnology, Azerbaijan National Academy of Sciences, AZ1143 Baku, Azerbaijan; 3Department of Chemistry, Science Faculty, Gazi University, Teknikokullar, Ankara 06500, Turkey; 4Department of Biological Sciences, Faculty of Science, King Abdulaziz University, Jeddah 21589, Saudi Arabia; 5Departments of Biochemistry & Cellular and Molecular Biology, Chemical and Biomolecular Engineering and Microbiology, University of Tennessee, Knoxville, TN 37996, USA; 6K.A. Timiryazev Institute of Plant Physiology, Russian Academy of Sciences, Botanicheskaya Street 35, 127276 Moscow, Russia; 7Faculty of Engineering and Natural Sciences, Bahcesehir University, Istanbul 34349, Turkey

**Keywords:** organometallic complexes, DCMU, OJIP curve, PSII-containing membranes, JIP-test, inhibition

## Abstract

Modern agricultural cultivation relies heavily on genetically modified plants that survive after exposure to herbicides that kill weeds. Despite this biotechnology, there is a growing need for new sustainable, environmentally friendly, and biodegradable herbicides. We developed a novel [CuL_2_]Br_2_ complex (L = bis{4H-1,3,5-triazino[2,1-b]benzothiazole-2-amine,4-(2-imidazole) that is active on PSII by inhibiting photosynthetic oxygen evolution on the micromolar level. [CuL_2_]Br_2_ reduces the F_V_ of PSII fluorescence. Artificial electron donors do not rescind the effect of [CuL_2_]Br_2_. The inhibitory mechanism of [CuL_2_]Br_2_ remains unclear. To explore this mechanism, we investigated the effect of [CuL_2_]Br_2_ in the presence/absence of the well-studied inhibitor DCMU on PSII-containing membranes by OJIP Chl fluorescence transient measurements. [CuL_2_]Br_2_ has two effects on Chl fluorescence transients: (1) a substantial decrease of the Chl fluorescence intensity throughout the entire kinetics, and (2) an auxiliary “diuron-like” effect. The initial decrease dominates and is observed both with and without DCMU. In contrast, the “diuron-like” effect is small and is observed only without DCMU. We propose that [CuL_2_]Br_2_ has two binding sites for PSII with different affinities. At the high-affinity site, [CuL_2_]Br_2_ produces effects similar to PSII reaction center inhibition, while at the low-affinity site, [CuL_2_]Br_2_ produces effects identical to those of DCMU. These results are compared with other PSII-specific classes of herbicides.

## 1. Introduction

Currently, new approaches in agricultural economics are being intensively developed and put into practice, which selectively endow economically significant (genetically modified) crops with survival when treated with modern chemical agents that effectively suppress the growth and development of unwanted plant species [1]. The other side of these approaches (no less significant than the creation of genetically modified plant species) are studies aimed at identifying and studying the mechanism(s) of action of the widest possible range and the largest number of different chemical agents—potential prototypes of new, more effective, more environmentally friendly, more bioremediable, etc. of compounds for suppressing the growth of weeds [2]. In addition, these detectable chemical agents may turn out to be new highly specific tools of scientific knowledge, acting as inhibitors, acceptors, or donors of an electron with special properties [3]. Many well-studied inhibitors of photosynthetic electron transfer, electron donors, and acceptors are already widely used in scientific research, for example, to isolate the desired site of the electron transport chain of photosynthesis [4,5] and even for practical purposes—for example, in modified natural systems for generating molecular hydrogen [5,6].

Photosynthesis, the only process of generating biomass on the planet from carbon dioxide and water due to the energy of solar radiation, is carried out by the photosynthetic apparatus (PA) of phototrophs, including higher plants. One of the main PA pigment–protein–lipid complexes is the PSII complex, which splits water into high-energy electrons and protons, and evolving oxygen. The structural and functional organization of PA, including PSII, has been considered in detail in numerous reviews [7]. Among all the complexes that make up PA, PSII, and, especially, the oxygen-evolving complex PSII, which oxidizes water, suffer most from the action of various stress factors, including chemical agents [8]. Suppressing the activity of PA, primarily its most vulnerable part, PSII, is the most effective way to block the growth of a plant. This approach underlies the action of the majority of herbicides, inhibitors of photosynthesis [5].

Among the studies of various chemical inhibitors of various kinds of biological activities, including the photosynthetic activity of the components of the electron transport chain of photosynthesis (but mainly PSII), studies are developing organic complexes [9,10,11,12], as well as complexes of organic ligands with semimetals (Sb, As, etc.) [13] and transition metals (Fe, Pb, Co, Ni, Cr, Zn) [3,14,15], including organometallic complexes based on copper (Cu(II)) [16,17].

Many of these metals and semimetals in their free form have a very high ability to enter into various reactions, but exhibit low solubility in hydrophobic media and, as a result, may not achieve their intended targets as plant growth inhibitors. This also applies to copper cations, which are efficiently bound by organic buffer solutions [18], e.g., diphenylcarbazide, an exogenous electron donor widely used in photosynthesis studies [19,20], that chemically interact in solution with sodium ascorbate [21], hydroxylamine [22], dithionite [23], quinones [24], ferricyanide [25].

The several effects of a new organometallic complex based on Cu(II) ions with organic ligand (L = bis{4H-1,3,5-triazino [2,1-b]benzothiazole-2-amine,4-(2-imidazole)}copper(II) bromide complex)—[CuL_2_]Br_2_ on photochemical activity in PSII-containing membranes have recently been described [26]. To relieve the reader of the need to refer to our previous publication, we present the structure of the ligand (A) and the Cu(II)-complex (B) here again in Figure 1.

It has been shown that [CuL_2_]Br_2_: (1) is not an artificial electron acceptor for PSII; (2) inhibits photosynthetic electron transfer measured as photoinduced oxygen evolution; (3) in steady-state measurements, diminished F_M_ values of the PSII chlorophyll fluorescence yield occur exclusively at the expense of F_V_ values, which do not recover by adding artificial electron donors. (4) This new compound presumably has no significant effect on the native state of PSII proteins nor on the interaction of PSI with PSII with efficacy, which is no more than 5–10% of the DCMU effect [26]. Based on obtained results, it was proposed that main effect on the PSII photochemical activity is probably due to the interaction of the inhibitory agent with reaction center (RC) leading to some conformational changes in its structure. Further studies using other methods are needed to obtain a more accurate answer.

Registration of the PSII fast chlorophyll fluorescence induction (OJIP-kinetic) and its comprehensive analysis (JIP-test) is a quick, non-invasive, non-destructive, reliable, sensitive, informative, and convenient research method with which information can be readily obtained concerning the state and functioning of practically every component of the donor and acceptor sides of the PSII RC as well as about all intermediates of the PA in intact samples under stress impacts, including effects of various inhibitors [27,28]. If several sites of inhibitor action are detected by JIP-testing of PSII, one may determine which of these sites of action (and/or effects) of the inhibitor are of a primary or secondary nature. It may be revealed by differences in the dependence of changes in the magnitude of different peaks of OJIP kinetics on the concentration of the added inhibitor, and/or by identifying additional peaks after appropriate normalization procedures and the subtraction of control kinetics OJIP from the OJIP-kinetics in the presence of the inhibitor [27,28].

Here, we used the JIP test to elucidate inhibitory impacts of [CuL_2_]Br_2_ on PSII-containing membranes in more detail. Furthermore, we studied the effects of [CuL_2_]Br_2_ on PSII-containing membranes in the presence of DCMU, a known inhibitor of electron transfer on the acceptor side of PSII, blocking the oxidation of the reduced primary electron acceptor Q_A_ (Q_A_^−^) by the plastoquinone molecules from the membrane pool [4,29,30]. Investigation of the effects of [CuL_2_]Br_2_ in the presence of DCMU is a constructive experimental approach, since it allows using DCMU to exclude the possible influence of the remaining (in the PSII-membranes after their isolation) molecules of plastoquinone electron acceptors (PQ-9), approximately two molecules of PQ-9 per PSII RC [31,32]. On the other hand, if it turns out that the effects of [CuL_2_]Br_2_ and DCMU manifest themselves dependently or independently of each other, then these data will make it possible to more clearly judge the possible site of action and/or binding of the new Cu(II)-complex, as was shown, for example, in a study evaluating the site of action and/or binding of perfluoroisopropyldinitrobenzene derivatives, inhibitors of the K-15 type [33], and a protein synthesis inhibitor, chloramphenicol [34].

## 2. Materials and Methods

### 2.1. Isolation of PSII-Containing Membranes

The oxygen-evolving PSII-containing membranes were isolated from the leaves of the greenhouse spinach (*Spinacia oleracea* L.), according to [35], with a little modification as in [36]. These PSII-containing membranes contain about 200 chlorophyll molecules per RC (per one molecule of photoactive pheophytin) [37] and, when illuminated with red light (λ ≥ 650 nm) of saturating intensity, they release oxygen in the presence of two artificial electron acceptors (0.1 mM 2,5-dichloro-*p*-benzoquinone and 1 mM K_3_Fe(CN)_6_) at a rate of 450–500 µmol O_2_ (mg Chl)^−1^ h^−1^. PSII-containing membranes suspended in medium (A) (50 mM MES-NaOH (pH 6.5), 300 mM sucrose, 15 mM NaCl) were stored at −80 °C. The concentration of total chlorophyll in PSII-containing membranes was determined by extraction with 96% (*v*/*v*) ethanol [38].

### 2.2. Fast Induction Kinetics of Chlorophyll Fluorescence

Fast induction kinetics of chlorophyll fluorescence associated with photoreduction of the PSII primary electron acceptor, plastoquinone Q_A_, were recorded using a MULTI-COLOR-PAM fluorimeter (Heinz Walz GmbH, Pfullingen, Germany) in a quartz cuvette (optical path length, 1 cm), at room temperature and constant stirring, after adaptation in the dark for at least 15 min. The final concentration of PSII-containing membranes in terms of chlorophyll was 4 µg mL^−1^. The conditions for measuring fast induction curves of chlorophyll fluorescence using this fluorimeter are described in detail in [39]. Each kinetic result represents are average of 5 independent experiments. The measurements were carried out as follows. A volume of the initial solution of PSII-containing membranes was prepared in medium containing: 50 mM MES–NaOH (pH 6.5), 300 mM sucrose, 15 mM NaCl, and then either an inhibitor solution or the same volume of solvent (in which this inhibitory agent was prepared) was added to an aliquot taken from this volume. This guaranteed the same chlorophyll concentration in all measurements. Based on the chlorophyll fluorescence fast induction curves, a number of fluorescence parameters of PSII chlorophyll were determined and/or calculated.

### 2.3. Spectrophotometric Measurements

The absorption spectra of the [CuL_2_]Br_2_ complex were recorded in a standard quartz cell (Hellma, Müllheim, Germany) with an optical path length of 10 mm on a two-beam Shimadzu spectrophotometer, model UV-1800 (Shimadzu UV-1800, Shimadzu Europa GmbH, Duisburg, Germany) in the wavelength range 200–700 nm (optical slit width 2 nm, write speed 2 nm s^−1^) at room temperature, in measurement medium used for OJIP kinetics. The concentration of the [CuL_2_]Br_2_ complex was 0.1 mM and corresponded to the maximum concentration used in all experiments.

### 2.4. Solutions of Inhibitory Agents

Stock solutions of (3-(3,4-dichlorophenyl)-1,1-dimethylurea, DCMU,) and [CuL_2_]Br_2_ and subsequent dilute solutions were prepared in dimethyl sulfoxide (DMSO). In all measurements, the final concentration of DMSO did not exceed 1%. In separate experiments, we have shown that DMSO at this concentration has no effect on either the intensity or the shape of the OJIP-kinetic.

## 3. Results

### 3.1. Original OJIP Kinetics

Figure 2 shows the original OJIP kinetics measured on PSII-containing membranes in the absence of other additives (control—kinetic 1) or in the presence of: 3.6 μM [CuL_2_]Br_2_ (kinetic 2); 14.5 μM [CuL_2_]Br_2_ (kinetic 3); 4 μM DCMU (kinetic 4); 3.6 μM [CuL_2_]Br_2_ + 4 μM DCMU (kinetic 5); 14.5 μM [CuL_2_]Br_2_ + 4 μM DCMU (kinetic 6). The F_V_/F_M_ ratio is a generally accepted, widely used measure characterizing the quantum yield of the primary photochemical reaction of PSII [28]. Based on the analysis of the original kinetics presented in Figure 2, it can be seen that all the studied agents and their combinations cause significant decreases in the Chl “a” fluorescence intensity, especially noticeable at the F_M_ level leading to a decrease in the variable fluorescence (F_V_). Furthermore, 3.6 μM [CuL_2_]Br_2_ (kinetics 2); 14.5 μM [CuL_2_]Br_2_ (kinetic 3); and 4 μM DCMU (kinetic 4) also induced insignificant increases in the F_0_ level (inset to Figure 2). In the presence of DCMU, such increases of F_0_ by [CuL_2_]Br_2_ are not evident. It is important to especially note that the decrease in the F_M_ value caused by both [CuL_2_]Br_2_ concentrations is at list in ten times more significant than the increase in the F_0_ value induced by [CuL_2_]Br_2_ without DCMU. An increase in the F_0_ level in the presence of DCMU has been repeatedly noted earlier on leaves [40], thylakoids [41], and PSII-containing membranes [42,43,44]. Both types of these changes (F_M_ and F_0_ levels) lead to a decrease in F_V_/F_M_ ratio.

### 3.2. Original OJIP Kinetics Normalized Relative to F_0_ (F_0.02ms_)

In order to make it easier to analyze and more clearly represent the possible changes caused by the agents added to the control (in the absence of other additives); in comparison with the control, normalization is carried out to the initial level of fluorescence F_0_, as a rule, by the value of F_20µs_ or F_50µs_ measured at 20 µs or 50 µs, respectively [28], but sometimes by the F_0_ value measured at time t = 0 [45,46]. In recent years, normalization to F_0.05ms_ has been favored, although normalization to F_0.02ms_ is acceptable and still quite common [47,48]. In addition, it is shown that the possible errors in the calculation of the parameters of the JIP test in the case when F_50µs_ is used as F_0_ is higher than for F_20µs_ and F_t_→0 [45].

The original OJIP-kinetics normalized relative to F_0_ are presented as F_t_ − F_0_ versus time in Figure 3 (where F_0_ is the fluorescence at time 0.02 ms; F_t_ is the fluorescence at time t). The analysis of the presented kinetics shows the following main properties of the obtained kinetics and their changes caused by the studied agents and their combinations.

Kinetics measured in the absence of additions (control) are completely identical to those recorded on PSII-containing membranes [42,43,44,49]. There is no peak **I** in the kinetic (plateau **J**–**I**), the main feature characterizing the kinetics of fast chlorophyll fluorescence induction measured on PSII-containing membranes [42,43,44,49] and therefore the kinetics will be designated below as OJP kinetics [42]. The absence of peak **I** (plateau **J**–**I**) in the OJP kinetics of PSII-containing membranes has been substantiated previously [42]. 

In the presence of both studied concentrations (3.6 μM and 14.5 μM) of [CuL_2_]Br_2_, a significant simultaneous almost synchronous decrease in the chlorophyll fluorescence intensity (F) is observed along the entire length of the OJP kinetics. The decrease also includes the F_J_ level (2–3 ms), and it is in greater extent in the presence of 14.5 μM [CuL_2_]Br_2_. The chlorophyll fluorescence decrease is especially significant at the F_M_ level—in the presence of 3.6 μM and 14.5 μM [CuL_2_]Br_2_ by 22% and 45%, respectively, kinetics 2 and 3, Table 1 compared with the control (kinetic 1). The F_M_ decrease is especially pronounced at 14.5 μM [CuL_2_]Br_2_ (kinetic 3). Let us designate these decreases in F (including F_J_ and F_M_) as described above as the “effect of [CuL_2_]Br_2_”.

Thus, these experimental data suggest that out of the total number of PSII-containing membranes, 22% and 45%, PSII-containing membranes (respectively, in the presence of 3.6 μM and 14.5 μM [CuL_2_]Br_2_) are no longer capable of photochemical reduction of the corresponding components of the acceptor side of PSII. This effect is a consequence of a certain suppressive effect of [CuL_2_]Br_2_ on the components providing either charge separation or the source of electrons from the components of the donor side of PSII, and onward can be excluded from further consideration because they no longer produce JIP kinetics due to the action of [CuL_2_]Br_2_. Therefore, the remainder of the total number of PSII-containing membranes that retained photochemical activity in the presence of 3.6 μM and 14.5 μM [CuL_2_]Br_2_, respectively, should be considered, namely 78% and 55%. And using these data, it will be possible to find out by what mechanisms [CuL_2_]Br_2_ disrupts the functioning of PSII and in what sequence these mechanisms function.

In addition, F_M_ is reduced in the presence of 4 μM DCMU and especially in the presence of its combinations with both concentrations of [CuL_2_]Br_2_ (Figure 3, Table 1). Moreover, in the case of a combination of 14.5 μM [CuL_2_]Br_2_ + 4 μM DCMU, an almost synchronous decrease in the chlorophyll fluorescence intensity (F) occurs along the entire length of the OJP kinetics, which are similar to described above.

In the presence of DCMU (without [CuL_2_]Br_2_), changes in the OJP kinetics characteristic of DCMU are observed (the so-called “DCMU effect”)—namely, an increase in the F_J_ peak to the so-called F_M_ peak (kinetic 4), the intensity of which is less than the F_M_ peak of control. The effects of DCMU have been repeatedly shown and explained previously by other authors [41,42,43,44,50]. In the presence of DCMU, all the amount of Q_A_ present in the sample is restored, which is expressed in an increase of the J peak to the highest possible level. At the same time, there is a decrease in the F_M_ value to a value that is 62% from the control F_M_. This decrease is due to the quenching of F by oxidized PQ-9 molecules [41,42,43,44,50]. A further decrease in the F_M_ intensity by above reason seems unlikely, since in a preliminary experiment, we showed that 4 μM DCMU inhibited the oxidation of all reduced Q_A_ molecules at the concentration of PSII-containing membranes we used.

Of particular interest and significance are the changes in OJP kinetics that occur in the presence of simultaneously both DCMU and [CuL_2_]Br_2_ (kinetics 5 and 6). Both kinetics are similar to the kinetics recorded in the presence of only DCMU (“DCMU effect” (kinetic 4)), but at the same time, the intensity of chlorophyll fluorescence decreases even more significantly over the entire OJP kinetics (“[CuL_2_]Br_2_) effect”). This decrease is especially evident in the case of 14.5 μM [CuL_2_]Br_2_)+ 4 μM DCMU (kinetic 6). The intensity of F at the F_M_ level decreases in the case of these combinations of inhibitors (3.6 μM [CuL_2_]Br_2_) + 4 μM DCMU) and (14.5 μM [CuL_2_]Br_2_) + 4 μM DCMU), by 50% and 66%, respectively, relative to the control F_M_.

In this case, in the presence of both combinations of DCMU with 3.6 μM and 14.5 μM [CuL_2_]Br_2_ (similar to situation without DCMU described above), there is for further research only part from the total number of PSII-containing membranes that retained photochemical activity, namely 50% and 34% in this case relative to F_M_ in the presence of 4 μM DCMU alone. In such case, in the presence of both combinations of DCMU with 3.6 μM and or 14.5 μM [CuL_2_]Br_2_, the remaining parts of the total number of PSII-membranes that retained photochemical activity, namely 50% and 34%, should be further considered.

Thus, DCMU induces a “DCMU effect” regardless of the presence of [CuL_2_]Br_2_. At the same time, [CuL_2_]Br_2_ effectively suppresses the F_M_ value both in the absence and in the presence of DCMU.

In the presence of DCMU, it is important to correctly estimate how much [CuL_2_]Br_2_ reduces the F_M_ value. Since a further decrease due to quenching of F by oxidized PQ-9 molecules remaining in PSII-membrane is unlikely, since oxidation of all available Q_A_ molecules is blocked by DCMU, then the observed decrease caused by both concentrations of [CuL_2_]Br_2_ in the presence of DCMU is based on another reason, and the percentage of decrease in F_M_ in this case should be calculated by taking as 100% the value of F_M_ measured in the presence of 4 μM DCMU. In this case, a further F_M_ reduction due to quenching of F by oxidized PQ-9 molecules remaining in PSII-containing membranes is unlikely, since oxidation of all available Q_A_ molecules is blocked by DCMU. Consequently, the observed F_M_ reduction caused by both concentrations of [CuL_2_]Br_2_ in the presence of DCMU is based on another reason. The percentage of decreased F_M_ reduction in this case should be calculated by taking as 100% the value of F_M_ measured in the presence of 4 μM DCMU. These calculated data are shown in Table 1 in parentheses and highlighted by asterisks. Comparing these data, we can see the following: in the absence of DCMU, both concentrations of [CuL_2_]Br_2_ suppress the F_M_ value by 22% and 45%, respectively, and in the presence of DCMU, by 19% and 44%, respectively. These values are fairly well comparable.

The revealed coincidence of the values of F_M_ decrease by [CuL_2_]Br_2_ in the presence of DCMU and without DCMU suggests that in both cases [CuL_2_]Br_2_ inhibits the activity of PSII-containing membranes by the same mechanism.

### 3.3. OJIP Kinetics Normalized Relative to F_20µs_ and F_M_

Many stresses, including high or low temperature stress; high light intensities; UV-B; inhibitors of PSII photochemical activity, etc., affect the photoinduced redox state of Q_A_, and this is reflected in the form of changes in the intensity of the F_J_ peak of OJIP kinetics and/or time to **J**-peak [40,41,51,52,53].

In Figure 3, it is not easy to understand how the intensity F changes at the level of peak **J** for almost every kinetic compared to the control, with the exception of kinetics 4 (4 μM DCMU) and 5 (3.6 μM [CuL_2_]Br_2_ + 4 μM DCMU) in which an increase in F**_J_** intensity is clearly shown. Normalization of the original OJP kinetics simultaneously relative to the value of F_0_ and the value of F_M_ makes it possible to reveal in more detail possible changes, including intermediate peaks, in the case of PSII-containing membranes—peak **J**. It was of interest to clarify more clearly how [CuL_2_]Br_2_ affects the properties of the **J** peak in the absence and the presence of DCMU.

Figure 4 shows the original OJP kinetics normalized relative to F_0.02ms_ and to F_M_. After such normalization, it became obvious that, in addition to the simultaneous decrease in the chlorophyll fluorescence intensity over the entire OJP kinetics (slightly at the F_J_ level (2–3 ms) and especially pronounced at the F_M_ level), which was clearly pronounced after normalization original OJP kinetics relative only to F_0.02ms_, now there are significant changes in OJP kinetics compared with the control in the presence of both concentrations of [CuL_2_]Br_2_, as well as their combinations with DCMU, which in this case became especially pronounced in the region of peak J (Figure 4).

From the data presented in Figure 4, it is evident that: (1) without DCMU in the presence of 3.6 μM [CuL_2_]Br_2_ (kinetic 2), the intensity of the **J** peak increases compared to the control (kinetic 1), but at a higher concentration of [CuL_2_]Br_2_ (14.5 μM) (kinetic 3), this effect, which is expressed in an increase in the **J** peak, already becomes significantly less; (2) in the case of a combination of 3.6 μM [CuL_2_]Br_2_ and 4 μM DCMU (kinetic 5), the J peak becomes a little bit higher compared to 4 μM DCMU (kinetic 4), however, at a higher concentration of [CuL_2_]Br_2_ (14.5 μM) in this combination inhibitors (kinetic 6), a significant decrease in the J peak is already observed.

Thus, in both above cases (namely in the absence and in the presence of DCMU), the differently directed effect on the F intensity of the **J** peak of these two concentrations of [CuL_2_]Br_2_ (3.6 μM and 14.5 μM) is clearly visible. It should emphasize that in the presence of DCMU the difference in the above effects between these concentrations is much greater. Despite the fact that after this normalization it is possible to identify additional changes in the OJP kinetics, nevertheless, in this case, these changes are not yet clearly expressed, and it is not possible to quantify the degree of these changes.

### 3.4. Comparison [CuL_2_]Br_2_ and DCMU Effects

Peak I is known to be absent in PSII-containing membranes [34,42,43,44,49]. The IP phase is directly related to PSI activity, while JI phase parallels the reduction of PQ pool [27,28].

Since there is no **I** peak in PSII-containing membranes, in order to more conveniently analyze and visualize possible changes at the level of the **J** peak, which are induced by the studied inhibitory agents and their combinations, we first double normalized the original kinetics relative to both the F_0_ level (F_0.02ms_) and to the level of finding the peak **I** (_30ms_), i.e., to the level F_30ms_, according to the formula:V_0I_ = (F_t_ − F_0_)/(F_I_ − F_0_),
in our case
V_0I_ = (F_t_ − F_0.02ms_)/(F_30ms_ − F_0.02ms_)

The resulting kinetics V_0I_ = (F_t_ − F_0.02ms_)/(F_30ms_ − F_0.02ms_) are shown in Figure 5A. Next, we subtracted the kinetic obtained in the absence of any additions (control) from the kinetics obtained in the presence of inhibitory agents, for each of the studied inhibitory agents and their combinations. The obtained difference kinetics W_0I_ = V_0I experiment_ − V_0I control_ are shown in Figure 5B.

It is known that DCMU blocks electron transfer from the reduced primary PSII electron acceptor, plastoquinone Q_A_, into the membrane pool of plastoquinones (PQ-9), competing with the PSII secondary electron acceptor, plastoquinone Q_B_ for the binding site on the so-called Q_B_ herbicide-binding site of the D1 protein. Therefore, in the presence of DCMU, the so-called “diuron effect” is observed, which is expressed on the original OJIP kinetics as a significant increase in fluorescence intensity at a level of 2–3 ms (peak **J**) compared to the control [40,41,51,52,53]. This can be especially clearly seen in the difference OJP-kinetics obtained by subtracting from OJP-kinetics measured in the presence of DCMU, the kinetics obtained in the absence of any additions (control) [52,53].

Preliminarily, for the conditions of our measurements (the concentration of PSII-containing membranes, expressed as the concentration of chlorophyll contained in them, is 4 μg mL^−1^), we found that the concentration of DCMU used by us (4 μM) causes practically maximal “diuron effect” on the chlorophyll fluorescence of PSII.

In addition, the use of higher concentrations of DCMU may be accompanied by the effects of DCMU on other sites of the PSII electron transport chain, as described earlier [54,55,56,57]. In order to quantify the “diuron effect” of other studied inhibitory agents or their combined use with DCMU, we evaluated the F_J_ values for the other difference OJP-kinetics (W_0I_ = V_0I exp_ − V_0I control_) presented in Figure 5B in % from that with DCMU. The magnitude of the “diuron effect” F_J_ measured in the presence of 4 μM DCMU, which indicates amount of reduced Q_A_ (Q_A_^−^), we took as 100%. And the effects of other supplements were evaluated in % relative to this effect of DCMU. The data obtained are presented in Table 2.

#### 3.4.1. Effects of [CuL_2_]Br_2_ in the Absence of DCMU

From the data presented in Table 2, it can be seen that in the absence of DCMU, low concentrations (3.6 μM) of [CuL_2_]Br_2_ cause a “diuron effect” of approximately 38% of that caused by DCMU. With an increase in the [CuL_2_]Br_2_ concentration to 14.5 µM, the “diuron effect” increases and is already 71% of the “diuron effect” caused by DCMU.

#### 3.4.2. Effects of [CuL_2_]Br_2_ in the Presence of DCMU

A completely different effect of the [CuL_2_]Br_2_ complex on the photochemical activity of PSII is observed when [CuL_2_]Br_2_ complex is added in the presence of DCMU. In this case, both concentrations (3.6 μM and 14.5 μM) of [CuL_2_]Br_2_ significantly reduced the “diuron effect” of DCMU from 100%, respectively, to 59% and to about 3%.

Thus, from the data presented in Table 2, it is obvious that in the absence of DCMU, the amount of reduced Q_A_ increases with increasing concentration of the [CuL_2_]Br_2_ complex. However, in the presence of DCMU, on the contrary, the amount of reduced Q_A_ decreases significantly with an increase in the concentration of the [CuL_2_]Br_2_.

We evaluated the potency of these effects of [CuL_2_]Br_2_, namely (1) the effect of increasing the amount of reduced Q_A_ in the absence of DCMU and (2) the effect of decreasing the amount of reduced Q_A_ in the presence of DCMU on the concentration of the [CuL_2_]Br_2_ complex from the slope of the corresponding fitted curves. It turned out that the second mechanism of action of the [CuL_2_]Br_2_ complex, which manifests itself in a decrease in the amount of reduced Q_A_ in the presence of DCMU, is about two times more effective than the first one, the accumulation of the amount of reduced Q_A_ in the absence of DCMU.

Based on the comparison of the positions of the J peaks on the time scale, it can be roughly assumed that in the presence of 4 μM DCMU, the time to reach the maximum value of the fluorescence intensity of the J peak (F_J_) on the difference kinetics W_OI_ = V_OI exp_ − V_OI control_ (Figure 5B), which characterizes the rate of Q_A_ reduction with increasing concentration of the [CuL_2_]Br_2_ complex, also increases—as can be seen when comparing the difference kinetics for (3.6 μM [CuL_2_]Br_2_ + 4 μM DCMU) and (14.5 μM [CuL_2_]Br_2_ + 4 μM DCMU). Moreover, this property of the [CuL_2_]Br_2_ complex to slow down the rate of photoinduced Q_A_ reduction even increases with an increase in its concentration with DCMU. This is evident when comparing the different kinetics of (3.6 µM [CuL_2_]Br_2_) and (3.6 µM [CuL_2_]Br_2_ + 4 µM DCMU), as well as (14.5 µM [CuL_2_]Br_2_) and (14.5 µM [CuL_2_]Br_2_ + 4 µM DCMU).

However, there is a more reliable way to quantify the rate of photoinduced Q_A_ reduction.

### 3.5. Estimation of the Rate of Photoinduced Reduction of Q_A_

Graphical or computational determination of the initial slope (M_0_) of the JIP kinetics makes it possible to estimate the rate of photoinduced reduction of Q_A_ and its changes as a result of various influences [53,58,59,60,61,62]. We have used both of these approaches.

The graphical data presented in Figure 6 allow you to see much more clearly what changes are induced by the studied agents in PSII photochemical reactions; the graphical approach is also used by other researchers [53,58,60,62]. In addition, we calculated the values of M_0_ using the corresponding formula M_0_ = 4 (F_300_ − F_0_)/(F_M_ − F_0_). The results of the calculations are presented in the Table 3.

It should be noted that the values (M_0_) determined on the basis of the data presented in Figure 6 almost coincide with those obtained as a result of calculations. However, since the “first” ones were determined as a result of approximating real values, the data obtained in the calculations should be considered more accurate.

From the data presented in the form of kinetics in Figure 6 and the corresponding values of M_0_ in the Table 3, it follows that in all variants in the presence of the studied agents (both concentrations of [CuL_2_]Br_2_ without DCMU, 4 μM DCMU, both combinations of [CuL_2_]Br_2_ with DCMU), the rate of accumulation of reduced Q_A_ (Q_A_^−^), compared to the control is above (Figure 6 and Table 3).

If we evaluate the rate of accumulation of reduced Q_A_ (Q_A_^−^), compared with DCMU, then in the absence of DCMU, both concentrations of [CuL_2_]Br_2_ increase the rate of accumulation of reduced Q_A_ (Q_A_^−^), compared with the control, as well as in the presence of DCMU alone, however, with a significantly lower efficiency compared to DCMU (46% and 32% of that of DCMU) (kinetics 4 and 5). Interestingly, in the presence of a lower concentration of [CuL_2_]Br_2_ (3.6 µM), this effect is greater (46%) compared with a higher concentration (14.5 µM) of this agent (32%), i.e., without DCMU, the ability to cause an increase in the rate of Q_A_ reduction decreases with increasing concentration of [CuL_2_]Br_2_.

In the presence of DCMU, [CuL_2_]Br_2_ also reduces the rate of accumulation of reduced Q_A_ (Q_A_^−^), respectively, to 69% and 56%, relative to DCMU (Table 3 of kinetics 2 and 3), and this effect of [CuL_2_]Br_2_ also increases with increasing concentration [CuL_2_]Br_2_.

We evaluated the effectiveness of the impact of [CuL_2_]Br_2_ on the rate of Q_A_ reduction in the absence and presence of DCMU by the slope of the approximated lines plotted using the corresponding experimental data from Table 3. It turned out that both in the absence of DCMU and with DCMU, the rate of Q_A_ reduction with increasing concentration of [CuL_2_]Br_2_ goes down. However, in the presence of DCMU, the rate of Q_A_ reduction with increasing concentration of [CuL_2_]Br_2_ decreases approximately three times faster than in the absence of DCMU.

### 3.6. Absorption Spectrum of [CuL_2_]Br_2_

It could be assumed that the observed simultaneous synchronous decrease in the intensity of the fast chlorophyll fluorescence induction curves and almost all its peaks in the presence of [CuL_2_]Br_2_, which enlarges with an increase in the concentration of this organometallic complex added to the measuring medium, could be due to (1) a decrease in the intensity of the measuring and/or acting light due to the screening effect—absorption of light quanta by [CuL_2_]Br_2_ or (2) a decrease in the intensity of fluorescence emitted by chlorophyll molecules due to its absorption by the molecules of the [CuL_2_]Br_2_—the effect of chlorophyll fluorescence reabsorption. To test this assumption, we studied the absorption spectrum of the [CuL_2_]Br_2_ complex. As shown in Figure 7, the [CuL_2_]Br_2_ complex has no absorption bands either in the region of wavelengths of both types of light or in the region of emission wavelengths of chlorophyll fluorescence. Therefore, the above assumption is erroneous.

## 4. Discussion

### 4.1. Main Inhibitory Impact of [CuL_2_]Br_2_ on OJP Transients

The strongest and, therefore, undoubtedly, the main effect of the studied complex [CuL_2_]Br_2_ on the photochemical activity of PSII-containing membranes is a total synchronous decrease in the F intensity along the entire JIP kinetics (Figure 2 and Figure 3, kinetics 2, 3 as well as Table 1). [CuL_2_]Br_2_ causes some changes in PSII when it becomes no longer capable of photoinduced Q_A_ reduction. Already at a concentration of 3.6 µM, [CuL_2_]Br_2_ is able to completely exclude 22% PSII-containing membranes from the total number of photochemically active PSII-containing membranes, but when at a concentration of 14.5 µM [CuL_2_]Br_2_ totally disables already 45% PSII-containing membranes (in the absence of DCMU). It is important to note that [CuL_2_]Br_2_ also demonstrates this effect on PSII in the presence of DCMU, with an efficiency that is well comparable to that estimated in the absence of DCMU (Figure 2 and Figure 3 (kinetics 5, 6) and Table 1, data highlighted in green). Based on these data, we can make an experimentally substantiated conclusion that the main effect of [CuL_2_]Br_2_ on PSII does not depend on DCMU. Fairly well-comparable values of F_M_ reduction by both concentrations of [CuL_2_]Br_2_ in the absence and presence of DCMU (Figure 3, Table 1) suggest that in both cases, PSII inhibition by the [CuL_2_]Br_2_ complex may be based on the same mechanism of action. This in turn suggests that [CuL_2_]Br_2_ does not need to bind to the DCMU binding site to exert this effect. The fact that [CuL_2_]Br_2_ suppresses F_M_ regardless of the presence of diuron suggests that the site of action and/or binding of [CuL_2_]Br_2_ on PSII is prior to the site of action and/or binding of diuron. Similarly, based on the obtained data about the independent manifestation of the effects of diuron and chloramphenicol on the OJIP kinetics of PSII-containing membranes, an experimentally substantiated conclusion was made that the site of action of chloramphenicol in PSII is located before the site of action of diuron [34].

What are the reasons for the revealed total synchronous decrease in the intensity F over the entire kinetics of JIP induced by [CuL_2_]Br_2_?

The corresponding decrease in the intensity F may be due to electron acceptance if this Cu(II)-complex acts as artificial electron acceptor. A similar decrease in fluorescence intensity is observed in the presence of known PSII electron acceptors, such as DCBQ [43,63]. The similar effect was observed in the case of chloramphenicol capable of effectively oxidizing pheophytin [34]. However, we have previously shown that [CuL_2_]Br_2_ is not an artificial electron acceptor, because it does not support photosynthetic oxygen evolution [26].

A significant simultaneous almost synchronous decrease in the fluorescence intensity of chlorophyll along the entire length of the OJP kinetics, especially at the F_M_ level, increasing with increasing concentration of [CuL_2_]Br_2_, which we designated as the “[CuL_2_]Br_2_ effect”, may be the result of a violation of the donor side or the reaction PSII center itself. Similar changes in OJIP kinetics were observed when the PSII donor side becomes non-functional [42,44]. However, artificial electron donors do not eliminate the inhibitory action of [CuL_2_]Br_2_ [26].

Based on experimental data obtained earlier [26], now we propose that [CuL_2_]Br_2_ probably acts directly on the reaction center of PSII, and it is concerning to its main impact. It was shown that single-walled carbon nanotubes (SWCNT) at concentration of 300 mg/L influence the fast chlorophyll fluorescence induction curve [64]. The effect is very similar to that of the [CuL_2_]Br_2_. In the presence of 300 mg/L SWCNT, a total decrease in fluorescence intensity occurs along the entire length of the OJIP kinetics, which is especially pronounced at the F_M_ level. It was shown that this effect of SWCNT is due to SWCNT inactivation of PSII RC [64]. Furthermore, earlier it was shown that Cu(II) aqua-ions act at the level of reaction centers of PSII [65].

### 4.2. Impact of [CuL_2_]Br_2_ on J and 0 Peaks

Realistically, the % of the main effect of [CuL_2_]Br_2_ on JIP-transients is probably induced rather by lower concentrations of [CuL_2_]Br_2_ discussed above. This assumption is based on the fact that the remaining part of the PSII-containing membranes in the presence of the indicated concentrations changed their properties if we compare the OJP kinetics in the presence of [CuL_2_]Br_2_ with the control OJP kinetics. It is more convenient to consider these auxiliary (not main) [CuL_2_]Br_2_ effects to separate variants in the presence and the absence of DCMU.

#### 4.2.1. Impact of [CuL_2_]Br_2_ on J and 0 Peaks in the Absence DCMU

[CuL_2_]Br_2_ causes a slight increase in F_0_ levels (Figure 2 and inset kinetics 2 and 3) like it is usually induced by DCMU and agents with similar inhibitory mechanism—stopping electron transfer from reduced Q_A_ onto the next mediator of electron transport chain. This is not observed in the presence of DCMU.

[CuL_2_]Br_2_ causes a slight increase in F_J_ levels (Figure 4 and Figure 5B). It is interesting that in these figures, at the first glance, there is opposite dependence of the [CuL_2_]Br_2_ effects on the FJ intensity. On Figure 4, at low [CuL_2_]Br_2_ concentration (3.6 μM), increasing of the FJ intensity seems more expressed than at 14.5 μM). Whereas on Figure 5B, the dependence is opposite.

In fact, a correct picture of the [CuL_2_]Br_2_ influence on the F_J_ level can be obtained by subtracting the control OJP kinetic doubly normalized relative to F_0_ and F_M_ from those in the presence of [CuL_2_]Br_2_, i.e., kinetics 2 and 4 shown on Figure 5B. This figure shows that the auxiliary effect of [CuL_2_]Br_2_ on the difference kinetics is similar to the effect of diuron, i.e., an increase in the amount of reduced Q_A_. This “diuron-like effect” increases with increasing concentration of [CuL_2_]Br_2_. Thus, in the absence of DCMU, both of the above results, namely, an increase in the level of F_0_ and F_J_, suggest that in the absence of DCMU, an auxiliary (not main) effect of [CuL_2_]Br_2_ is that [CuL_2_]Br_2_ acts like a DCMU, but with less efficiency than DCMU.

#### 4.2.2. Impact of [CuL_2_]Br_2_ on J and 0 Peaks in the Presence DCMU

Figure 4 shows that in the presence of DCMU, both concentrations of [CuL_2_]Br_2_ seem to cause an increase in the J level comparable to that induced by DCMU (kinetics 5 and 6). However in fact, it must be taken into account that this increase in J is a joint effect of DCMU and [CuL_2_]Br_2_. The real picture of the influence of both [CuL_2_]Br_2_ concentrations on the magnitude of the J peak in the presence of DCMU is clear only on the difference kinetics (Figure 5B of kinetics 3 and 5). Figure 5B shows that both [CuL_2_]Br_2_ concentrations in fact decrease the DCMU effect, and the decrease is more at higher [CuL_2_]Br_2_ concentration (respectively, to 59% and to about 3% relative to 100% DCMU effect) (Table 2).

It could be assumed that [CuL_2_]Br_2_ can displace DCMU from its binding site, and this results in the decrease in the effect of DCMU. However, in this case, kinetics 3 and 5 would not be observed, but kinetics 2 and 4, because in the absence of DCMU, [CuL_2_]Br_2_ causes such auxiliary effects (the so-called “diuron effect”). However, this is in fact not the case. This means that in the presence of DCMU, [CuL_2_]Br_2_ causes some changes in PSII when it is no longer capable of photoinduced Q_A_ reduction even in the presence of DCMU, an effect similar to the main effect of [CuL_2_]Br_2_.

### 4.3. The Rate of Photoinduced Reduction of Q_A_

The fact that in the absence of DCMU simultaneously with the main effect [CuL_2_]Br_2_ appears to have a much less effective auxiliary effect is also evidenced by the fact that the M_0_ values reflecting the rate of photoinduced Q_A_ reduction in the presence of this complex are higher compared to the control (Figure 6 kinetics 4 and 5, Table 3). That higher M_0_ values are the result of an auxiliary effect of [CuL_2_]Br_2_ is evidenced by the fact that the ability to cause an increase in the rate of Q_A_ reduction decreases with increasing concentration of [CuL_2_]Br_2_. This trend is also observed in the presence of DCMU. The evidence that in the presence of DCMU, the rate of Q_A_ reduction with increasing concentration of [CuL_2_]Br_2_ decreases approximately three times faster than in the absence of DCMU also strongly suggests that in this case the ancillary effect of [CuL_2_]Br_2_ becomes the main one.

It could be assumed that the revealed decrease in the F intensity of OJP kinetics caused by [CuL_2_]Br_2_ is the result of the physical or functional separation of the antenna from the RC [66]. In this case, an increase in the F_0_ level can serve as a fairly reliable indication of this effect [66]. In our case, in the absence of DCMU, [CuL_2_]Br_2_ does not cause any increase in the F_0_ level, and it is obvious that its effect on PSII is not associated with the separation of the antenna from the RC.

It could be assumed that the decrease in the intensity of the chlorophyll fluorescence through whole OJP kinetic (but not F_0_ level) in the presence of our exogenous agent is the result of quenching of the chlorophyll fluorescence. It has been previously shown in thylakoids that DCBQ can act as an artificial electron acceptor and as a chlorophyll fluorescence quencher [63]. In this case, in addition to a decrease in the chlorophyll fluorescence intensity over the entire OJP kinetics, as a result of electron acceptance and an increase in the photochemistry rate, a decrease in the F_0_ level is also observed as a result of F antenna quenching. Moreover, it is important that the decrease in F_0_ caused by DCBQ is observed even in the presence of DCMU [63]. In our studies in the presence of DCMU, [CuL_2_]Br_2_ does not cause any decrease in the F_0_ value and, therefore, it is not a chlorophyll fluorescence quencher (Figure 2 and Figure 3 kinetics 5, 6).

The fact that the [CuL_2_]Br_2_ complex has no absorption bands either in the region of wavelengths of both types of light, or in the region of chlorophyll fluorescence emission wavelengths (Figure 7) gives grounds to suggest that a decrease in the chlorophyll fluorescence intensity of PSII-containing membranes in the presence of the [CuL_2_]Br_2_ complex could be due to its screening effect on the measuring and/or acting light. Another reason—the reabsorption of chlorophyll fluorescence by molecules of the [CuL_2_]Br_2_ could be excluded.

## 5. Conclusions

Based on the results obtained, we can assume:

(1) The main (dominating in terms of the degree of inhibition of PSII activity) effect of [CuL_2_]Br_2_ on PSII is probably associated with inhibition of the activity of the PSII RC. (2) The manifestation of auxiliary effects of [CuL_2_]Br_2_ on PSII is determined by the presence of DCMU. In the absence of DCMU, i.e., when the DCMU binding site is free, some part of [CuL_2_]Br_2_ that is involved in the induction of an auxiliary effect on PSII causes a “diuron-like” effect—an increase in the level of F_0_ and F_J_, i.e., blocks electron transfer from the reduced Q_A_ into the electron transport chain, but with less efficiency compared to DCMU. However, in the presence of DCMU, i.e., when the DCMU binding site is occupied by DCMU, this part of [CuL_2_]Br_2_, which is involved in the induction of the auxiliary effect on PSII, participates in the induction of the main effect, i.e., a total decrease in the intensity F over the entire OJP kinetics. Thus, it can be assumed that in PSII-containing membranes, there are two binding sites for [CuL_2_]Br_2_ with different affinities for [CuL_2_]Br_2_. At the high affinity site, [CuL_2_]Br_2_ produces effects similar inhibition of the PSII RC activity, while at the low affinity site, [CuL_2_]Br_2_ produces effects similar to those of DCMU. The data obtained can be useful in the development of promising herbicides for use in agricultural economics.

## Figures and Tables

**Figure 1 biomolecules-13-01058-f001:**
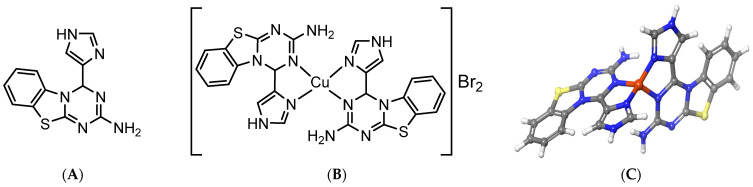
Ligand (L), 4H-1,3,5-triazino [2,1-b]benzothiazole-2-amine,4-(2-imidazole) (**A**), structure of [Cu(II)L_2_]Br_2_ complex (**B**), optimized structure of cationic copper(II) complex (**C**) [26]. Structure of the ligand (**A**) and the Cu(II)-complex (**B**) are given in Figure 1. The Cu(II)-complex is a [CuL_2_]^2+^ mononuclear cationic complex, with two bromide counterions to achieve neutrality, based on MS spectrum corresponding to [CuL_2_]^2^ cation, a 1:2 electrolyte matching molar conductivity measurement, and elemental analysis values. The neutral bidentate ligand is bound to a copper(II) atom with an imidazole nitrogen atom and benzothiazol nitrogen atom. Geometrical optimization calculation with DFT/B3LYP/6-31G(d,p) method showed that it has distorted tetrahedral geometries around Cu(II) atom.

**Figure 2 biomolecules-13-01058-f002:**
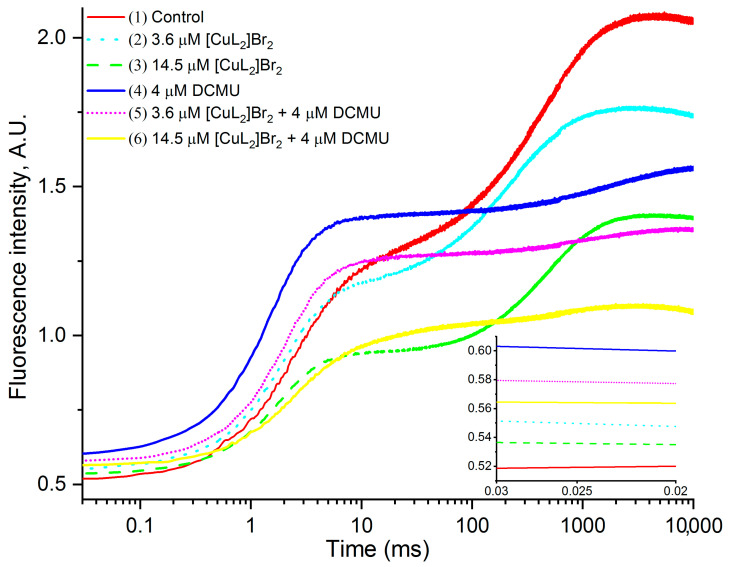
Original OJIP kinetics without any normalization, measured on PSII-containing membranes in the absence of other additions (control—kinetic 1) or in the presence of: 3.6 μM [CuL_2_]Br_2_ (kinetic 2); 14.5 μM [CuL_2_]Br_2_ (kinetic 3); 4 μM DCMU (kinetic 4); 3.6 μM [CuL_2_]Br_2_ + 4 μM DCMU (kinetic 5); 14.5 μM [CuL_2_]Br_2_ + 4 μM DCMU (kinetic 6). For clarity, the inset shows the initial positions of each kinetics.

**Figure 3 biomolecules-13-01058-f003:**
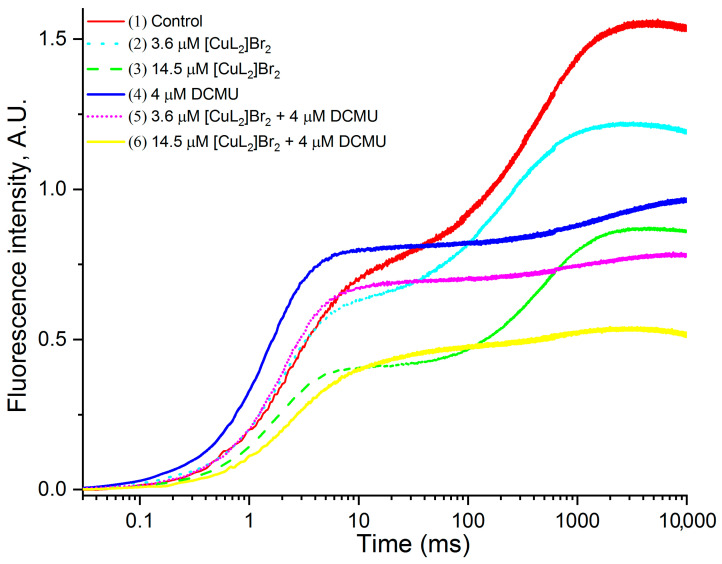
OJIP kinetics normalized relative to F_0.02ms_, measured on PSII-containing membranes in the absence of other additives (control—kinetic 1) or in the presence of: 3.6 μM [CuL_2_]Br_2_ (kinetic 2); 14.5 μM [CuL_2_]Br_2_ (kinetic 3); 4 μM DCMU (kinetic 4); 3.6 μM [CuL_2_]Br_2_ + 4 μM DCMU (kinetic 5); 14.5 μM [CuL_2_]Br_2_ + 4 μM DCMU (kinetic 6).

**Figure 4 biomolecules-13-01058-f004:**
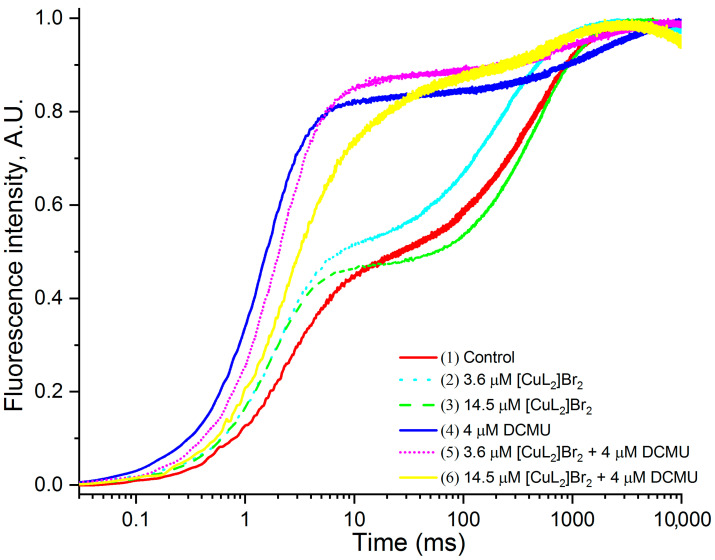
OJP kinetics normalized relative to F_0.02ms_ and F_M_, measured on PSII-containing membranes in the absence of other additives (control—kinetic 1) or in the presence of: 3.6 μM [CuL_2_]Br_2_ (kinetic 2); 14.5 μM [CuL_2_]Br_2_ (kinetic 3); 4 μM DCMU (kinetic 4); 3.6 μM [CuL_2_]Br_2_ + 4 μM DCMU (kinetic 5); 14.5 μM [CuL_2_]Br_2_ + 4 μM DCMU (kinetic 6).

**Figure 5 biomolecules-13-01058-f005:**
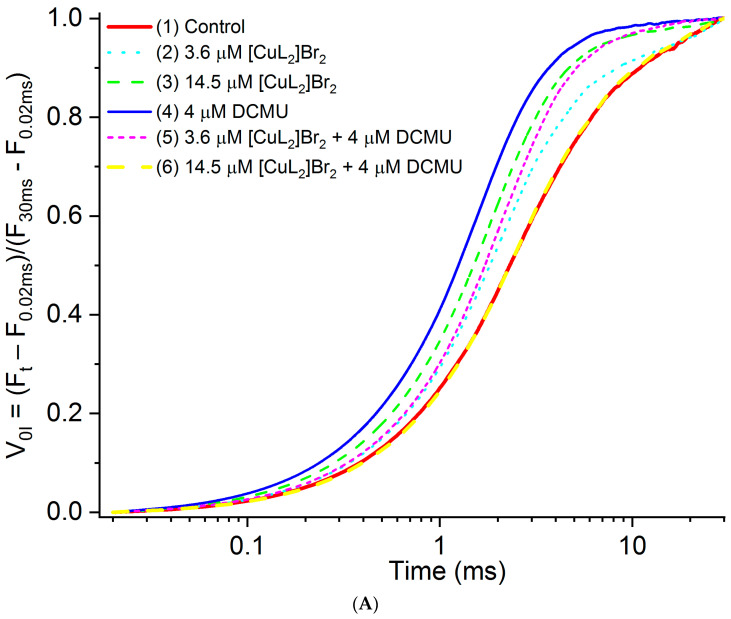
(**A**) OJP kinetics double normalized relative to F_0.02ms_ and F_30ms_, measured on PSII-containing membranes (V_OI_ = (F_t_ − F_0.02ms_)/(F_30ms_ − F_0.02ms_)) in the absence of other additives (control—kinetic 1) or in the presence of: 3.6 μM [CuL_2_]Br_2_ (kinetic 2); 14.5 μM [CuL_2_]Br_2_ (kinetic 3); 4 μM DCMU (kinetic 4); 3.6 μM [CuL_2_]Br_2_ + 4 μM DCMU (kinetic 5); 14.5 μM [CuL_2_]Br_2_ + 4 μM DCMU (kinetic 6). (**B**) The difference OJP-kinetics (W_OI_ = V_OI exp_ − V_OI cont_) plotted in the 0.02–30 ms time range, and obtained by subtraction of the control OJP-transients (in the absence of any inhibitors) from the OJP-transients of the PSII-membranes in the presence of the both concentrations of [CuL_2_]Br_2_ and their combinations with DCMU.

**Figure 6 biomolecules-13-01058-f006:**
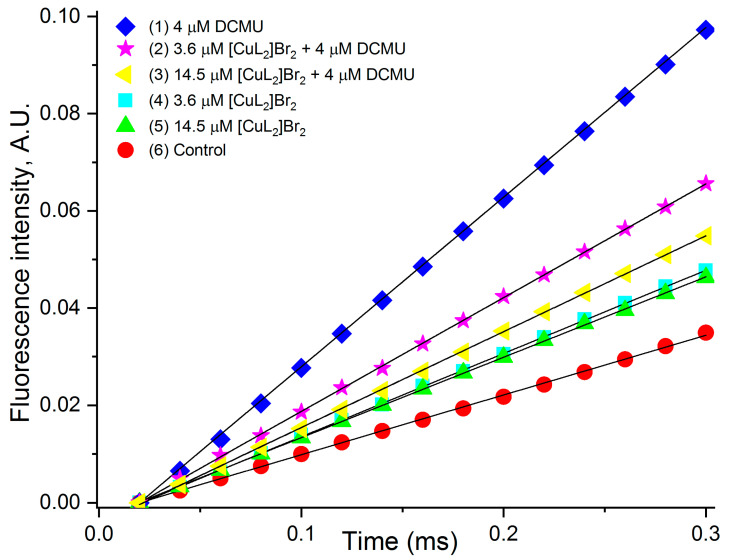
Initial sections of OJP kinetics normalized relative to F_0_ (F_0.02ms_) and maximum fluorescence F_M_, V_t_ = (F_t_ − F_0_)/(F_M_ − F_0_) = f(t), on a linear time scale from 0.02 ms to 0.3 ms in the absence and presence of the inhibitors indicated in the figure or their combinations. The black color shows the straight lines obtained by fitting the initial portion of each kinetic used to determine the M_0_ values.

**Figure 7 biomolecules-13-01058-f007:**
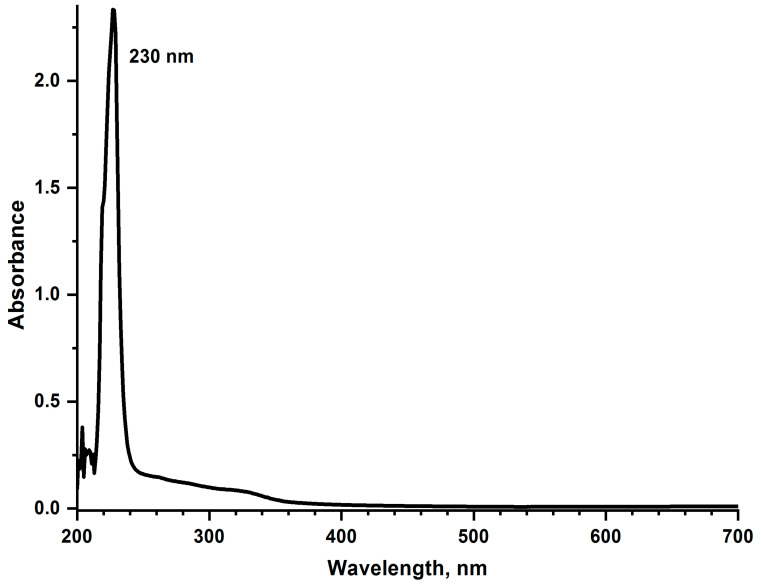
Absorption spectrum of [CuL_2_]Br_2_ at a concentration of 0.1 mM in the range of 200–700 nm, in the medium for measuring OJIP-kinetics at room temperature.

**Table 1 biomolecules-13-01058-t001:** Maximal recorded fluorescence intensities, at the peak P of OJP kinetics (F_M_ values) expressed as % of control as well as F_M_ decreases as % of control determined for both concentrations of [CuL_2_]Br_2_ and for their combination with DCMU.

Variants	F_M_, % of Control	F_M_ Decreases, % of Control
Control	100	0
3.6 µM [CuL_2_]Br_2_	78	22
14.5 µM [CuL_2_]Br_2_	55	45
4 µM DCMU	62 (100)	38 (0)
3.6 µM [CuL_2_]Br_2_ + 4 µM DCMU	50 (81)	50 (19)
14.5 µM [CuL_2_]Br_2_ + 4 µM DCMU	34 (56)	66 (44)

**Table 2 biomolecules-13-01058-t002:** Fluorescence intensity at the J-step (2 ms) of the difference OJP-kinetics (W_OI_ = V_OI exp_ − V_OI cont_)—the F_J_ values (so-called “diuron effect”) of the both concentrations of [CuL_2_]Br_2_ and their combinations with DCMU expressed as % of F_J_ with DCMU.

Variants	F_J_, % of F_J_ with DCMU
4 µM DCMU	100
3.6 µM M [CuL_2_]Br_2_	37.9 ± 0.3
14.5 µM M [CuL_2_]Br_2_	71.3 ± 0.3
3.6 µM [CuL_2_]Br_2_ + 4 µM DCMU	59.0 ± 0.3
14.5 µM [CuL_2_]Br_2_ +4 µM DCMU	2.9 ± 0.3

**Table 3 biomolecules-13-01058-t003:** Calculated values of M_0_—initial slope of the fluorescence transient normalized on the maximal variable fluorescence F_V_ expressed as % of control and % of DCMU.

Variants	M_0_, % of Control	M_0_, % of DCMU
4 µM DCMU	290	100
3.6 µM [CuL_2_]Br_2_ + 4 µM DCMU	199	69
14.5 µM [CuL_2_]Br_2_ + 4 µM DCMU	161	56
3.6 µM [CuL_2_]Br_2_	146	50
14.5 µM [CuL_2_]Br_2_	132	46
Control	100	35

## Data Availability

The data presented in this study are available on request from the corresponding authors.

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
