# Peer review of "Probing the Influence of Novel Organometallic Copper(II) Complexes on Spinach PSII Photochemistry Using OJIP Fluorescence Transient Measurements"

_biomolecules, 2023, doi:10.3390/biom13071058_

Round 1

Reviewer 1 Report

The main aim of the manuscript is to understand the herbicide effect on PSII. They have used novel organometallic copper(II) complexes to the inhibition effect on PSII. The manuscript is well designed and executed.

Before it consider for the publication I have a few queries.

1. The effect of DCMU and Copper (II) molecules behaves differently. As you see the results of the DCMU JI phase are almost missing. Whereas CuL2]Br2 is still inducing the IP phase. To, me electron transport is still occurring through CuL2]Br2. How do you defend?

2. The language needs to polish as I felt a bit difficult to understand in many places. 

Author Response

Reviewer 1

The main aim of the manuscript is to understand the herbicide effect on PSII. They have used novel organometallic copper(II) complexes to the inhibition effect on PSII. The manuscript is well designed and executed.

Before it consider for the publication I have a few queries.

 Comments:

  1. The effect of DCMU and Copper (II) molecules behaves differently. As you see the results of the DCMU JI phase are almost missing. Whereas CuL2]Br2 is still inducing the IP phase. To, me electron transport is still occurring through CuL2]Br2. How do you defend?

Response: It is done.

Thanks.

We sincerely apologize for inadvertently misleading you. Sentence: “To explore this mechanism, we investigated the effect of [CuL2]Br2 in the presence/absence of the well-studied inhibitor DCMU on PSII-containing thylakoid membranes by OJIP fluorescence transient measurements” (In Abstract) contains unfortunate misspelling.

It must be as: “To explore this mechanism, we investigated the effect of [CuL2]Br2 in the presence/absence of the well-studied inhibitor DCMU on PSII-containing membranes by OJIP fluorescence transient measurements”

All results (presented in manuscript) were obtained on PSII-containing membranes like BBY-particle not on thylakoids.

Furthermore, in legends to each figure as well as everywhere in text we indicate PSII-containing membranes not thylakoids.

It is known, and we discuss this in the text of the manuscript (with references) lines 234-240, that the OJIP kinetics of PSII-containing membranes do not contain an I peak. Text from manuscript (line 234-240) is below:

  1. Kinetic measured in the absence of additions (control) are completely identical to those recorded on PSII-containing membranes [42–44,49]. There is no peak I in the kinetic (plateau J-I), the main feature characterizing the kinetics of fast chlorophyll fluorescence induction measured on PSII-containing membranes [42–44,49] and therefore the kinetics will be designated below as OJP kinetics [42]. The absence of peak I (plateau J-I) in the OJP kinetics of PSII-containing membranes has been substantiated previously [42].

We sincerely apologize for inadvertently misleading you in Abstract.

We present an additional figure showing the OJIP kinetics measured on thylakoids and on PSII-containing membranes like BBY-particle.

 It is done.

  1. The language needs to polish as I felt a bit difficult to understand in many places.

Response:

Thanks.

The language has been corrected by native English speakers.

Reviewer 2 Report

The current manuscript aims to reveal an inhibitory mechanism of the organometallic complex [CuL2]Br2 – a novel inhibitor of PSII. The authors compared the influence of DCMU and the investigated compound on the chlorophyll fluorescence transients of PSII membranes (OJIP-kinetic). After normalising the kinetic curves, a series of calculations was performed and the authors concluded that [CuL2]Br2 can bind to PSII at two sites with different affinity. High-affinity binding caused an effect similar to inhibition of the PSII reaction centre, but binding to a side with low affinity resulted in effects similar to those which DCMU gives. Although the results presented are interesting, in my opinion, they can be published after a revision. 

Major points: 

  1. In Conclusions, the authors suggest that [CuL2]Br2 may bind to PSII at the same place as DCMU, but with less efficiency. Is it possible to determine with more precision the affinity of [CuL2]Br2 and DCMU to PSII?
  2. There are errors in the numbering of chapters and subsections, e.g. the chapter Materials and methods has number 2, and the subsection numbers of this chapter start with 4. Similar mistakes are made in the Results and Discussion chapters.
  3. There is inconsequence in citing the literature. On page 6, lines 226-234, reference numbers are places in the brackets ant they are followed by authors names and years of publications.
  4. There are some unfinished sentences, for example, page 4, line 162.
  5. On page 12, line 430 it is written that 8 µM [CuL2]Br2 was used, but in the table cited (Table 2) and in the figures, different concentrations of this compound are labelled.
  6. On page 17, line 601 the authors write that increasing of I peak intensity was observed but, as was pointed out before (page 6, line 228), I peak was not observed in the kinetic curves.
  7. The language of the manuscript needs to be improved.

Author Response

Reviewer 2

The current manuscript aims to reveal an inhibitory mechanism of the organometallic complex [CuL2]Br2 – a novel inhibitor of PSII. The authors compared the influence of DCMU and the investigated compound on the chlorophyll fluorescence transients of PSII membranes (OJIP-kinetic). After normalising the kinetic curves, a series of calculations was performed and the authors concluded that [CuL2]Br2 can bind to PSII at two sites with different affinity. High-affinity binding caused an effect similar to inhibition of the PSII reaction centre, but binding to a side with low affinity resulted in effects similar to those which DCMU gives. Although the results presented are interesting, in my opinion, they can be published after a revision.

Major points:

Comments:

  1. In Conclusions, the authors suggest that [CuL2]Br2 may bind to PSII at the same place as DCMU, but with less efficiency. Is it possible to determine with more precision the affinity of [CuL2]Br2 and DCMU to PSII?

Response: It is done.

Thanks.

Evidently main effect of [CuL2]Br2 is more important and interesting. Nevertheless we also plan to design and conduct these studies as a competition between DCMU and [CuL2]Br2, for example. The problem is that this "diuron-like" effect of [CuL2]Br2 is not the main but auxiliary, rather weak, it appears only in the absence of DCMU, and all manifestations of this "diuron-like" effect of [CuL2]Br2 are totally similar to all manifestations of DCMU.

Comments:

  1. There are errors in the numbering of chapters and subsections, e.g. the chapter Materials and methods has number 2, and the subsection numbers of this chapter start with 4. Similar mistakes are made in the Results and Discussion chapters.

Response: It is done.

Thanks.

The numbering of chapters and subsections was totally corrected. Corrected numbering of chapters and subsections is typed by red font and highlighted by green.

Comments.

  1. There is inconsequence in citing the literature. On page 6, lines 226-234, reference numbers are places in the brackets ant they are followed by authors names and years of publications.

Response: It is done.

Thanks.

The references as authors names and years of publications in the brackets were totally removed.

Comments:

  1. There are some unfinished sentences, for example, page 4, line 162.

Response: It is done.

Thanks.

Unfinished sentence: “They are:” was removed.

Comments:

  1. On page 12, line 430 it is written that 8 µM [CuL2]Br2 was used, but in the table cited (Table 2) and in the figures, different concentrations of this compound are labelled.

Response: It is done.

Thanks.

Instead - by a factor of 2 to 8 μM

Now - to 14,5 µМ

Comments:

  1. On page 17, line 601 the authors write that increasing of I peak intensity was observed but, as was pointed out before (page 6, line 228), I peak was not observed in the kinetic curves.

Response: It is done.

Thanks.

It is our misprint. It was corrected.

increasing of the FI intensity

increasing of the FJ intensity

Comments:

  1. The language of the manuscript needs to be improved.

Response: It is done.

Thanks.

The language has been corrected by native English speakers.

Reviewer 3 Report

The manuscript titled, “Probing the influence of…transient measurements” authored by Zharmukhamedov et al. explores the inhibitory effect of a novel [CuL2]Br2 complex in the isolated thylakoid membranes from Spinach through fluorescence induction assay (OJIP). The manuscript is closely related to their previously published work (DOI: 10.3390/cells11172680) and in some way repetitive. Below I highlight some major concerns that I have regarding the current manuscript.

General Comments

1.     The manuscript needs proper proofreading. Throughout the text, there are shortcomings that the authors might have avoided by careful proofreading. I will highlight certain sections here as examples, and I cannot reiterate enough that these are not the only shortcomings.

a.     Incomplete sentences: There are many instances in the manuscript where it seems like the authors want to say more but have not. Below I write two such examples that I could immediately notice. The authors need to read the manuscript word-by-word to rectify such mistakes.

                                               i.     Line 38- “the entire OJIP measurement, and 2) an auxiliary "diuron-like".” What the authors wish to say is unclear when they abruptly end the sentence with “auxiliary “diuron-like”.”

                                             ii.     Section 4.2, Line 162- “determined and/or calculated. They are:”. Another example of an incomplete sentence.

                                           iii.     Line 370: “Peak I is assumed reflects”. Reflects what? What is assumed?

b.     Grammatical and spelling errors: The text is full of grammatical errors that could have been avoided. Again the words mentioned below are few examples and are not the only instances.

                                               i.     Line 40: proposed should be propose.

                                             ii.     Line 71: “in this case, a weed”, What are the authors referring to when they write “in this case a weed” immediately followed by a paragraph that deals with photosynthetic apparatus, PSII, PSI, etc.

                                           iii.     Line 307: decrease is misspelled

c.     Inconsistency: The authors need to be consistent with defined abbreviations. For example, At many places, the concentration is represented as 3,6 μM (Line 188; including figures), and at other places, they write 3.6 μM (Line 253). Photosystem II is written as PS2 or PSII (Line 67). Photosynthetic apparatus has been defined as PA (Line62-63), but authors spell it out in Line 115. Line 104 describes Fv as variables F(ΔF), what is signified by ΔF or F per se? Section 4.4- DMSO is referred to as DMCO. Sections and sub-sections are numbered incorrectly; Section 2 has sub-sections numbered 4.1 etc. while Section 3 has sub-sections numbered as 2.1 etc. Section 4 has sub-sections numbered 3.1 etc.

2.     Factual incorrectness or missing references:

a.     Line 130-131: The authors state 2-3 molecules of PQ-9 per one reaction center and provide references 31 and 32 to back the statement. The references are from the years 1998 and 2000. Since the year 2000, many crystal structures of PSII from various organisms have been resolved, including spinach. To my knowledge only one structure from Thermosynechococcus elongatus (Guskov et al., 2009) shows the presence of a third PQ-9. The third PQ-9 was later proposed as a transient PQ-9 either leaving or entering PSII (Van Eerden et al., 2017). No high-resolution PSII structure since then has identified more than two PQ-9 in PSII. The authors should refer to more recent studies before making such a claim.

b.     Section 4.1, Line 139-140: I don’t know what is PSII-membrane. I would understand writing PSII-containing membrane or thylakoid membrane. What is greenhouse spinach?

Specific comments

1.     Figure 1 is an exact copy of Figure 1 from the previously published work of the authors (DOI: 10.3390/cells11172680). Moreover, the authors have failed to refer to Figure 1 in the text.

2.     Methods: Were the experiments carried out in one day, or the thylakoid membranes were repeatedly frozen and thawed? As the authors may appreciate that multiple freeze-thaw cycles may compromise the integrity of protein complexes.

a.     Section 4.1 does not describe the instrument or the temperature or the light source, and the intensity or optical filter used for the oxygen evolution assay. This information can be critical in determining the overall outcome of the oxygen evolution assay. After reading the section, I was expecting to see the oxygen evolution data but realized that this was done to check PSII integrity in the samples. Authors may want to remove the parts describing oxygen evolution and refer to their previous work to avoid confusion.

b.     Section 4.2 lacks a description of the dilution methods or buffer or incubation times with inhibitors. Looking at Figure 2, I understand that the buffer possibly contained DCBQ and potassium hexaferricyanide (I might be wrong in assuming so). Authors may want to explain a bit more. The fluorescence rise between J and P inflection points are dependent on the forward electron transport through PQ-9. Since thylakoid membrane preparation lacks free PQ-9, there should be a slow continuous rise till the fluorescence reaches P with an almost negligible J peak. A clear description of the PAM fluorimeter protocol could help the readers.

3.     A major problem I have with the results is the lack of repeats. The authors should mention if the repeats were performed or not. The data presented in Figure 3 is probably normalized as (Ft-F0) and not (Ft/F0) as mentioned in Line 222. If my math is correct normalizing by using Ft/F0 should result in a graph that starts at 1 and not 0 on the Y-axis. In lines describing V0I (Lines 379 and 380) F0.02MC and F0.02ms is confusing, again the authors need to be consistent. Figure 5B legend says data is presented in the time range of 0.02-3 ms, but the figure shows up to 30 ms. Comparing the data presented in Figure 5B and Table 2, the J-step is fixed at 2 ms; however, examining Figure 5B shows that the J peak is shifted beyond 2.5 ms for lines 3 and 4, making the percent changes presented in Table 2 incorrect.

4.     The discussion section is redundant, unnecessarily lengthy, and ignores the impact of Cu2+ on photosystem II. How about Cu substituting the active metal ions like Mg and Mn at higher concentrations? All discussion is focused on identifying the site for [CuL2]Br2 with no explanation given for the decrease in FM when the organo-metallic complex is present. I will leave the topic to the imagination of the authors to explain the decline of FM.

5.     The Bibliography section needs work. Reference 45 and 46-Why are the authors’ names capitalized? References 51 and 52 have texts that are probably not part of the title.  

6.     Funding section says “Figure 6 was obtained under the state contract…” Figure 6 in the text is experimental data. Authors need to correct it.

Author Response

Reviewer 3

General Comments

Comments:

  1. The manuscript needs proper proofreading. Throughout the text, there are shortcomings that the authors might have avoided by careful proofreading. I will highlight certain sections here as examples, and I cannot reiterate enough that these are not the only shortcomings.

Response: It is done.

Thanks.

It is done.

Comments:

  1. Incomplete sentences: There are many instances in the manuscript where it seems like the authors want to say more but have not. Below I write two such examples that I could immediately notice. The authors need to read the manuscript word-by-word to rectify such mistakes.

Response: It is done.

Thanks.

We carefully have read the manuscript word-by-word to rectify such mistakes

Comments:

  1. Line 38- “the entire OJIP measurement, and 2) an auxiliary "diuron-like".” What the authors wish to say is unclear when they abruptly end the sentence with “auxiliary “diuron-like”.”

Response: It is done.

Thanks.

[CuL2]Br2 has two effects on OJIP transients measurements: 1) a substantial decrease of the overall Chl fluorescence intensity over the entire OJIP measurement, and 2) an auxiliary "diuron-like" effect.

[CuL2]Br2 has two effects on OJIP transients: 1) a substantial decrease of the overall Chl fluorescence intensity, and 2) an auxiliary "diuron-like" effect.

Comments:

  1. Section 4.2, Line 162- “determined and/or calculated. They are:”. Another example of an incomplete sentence.

Response: It is done.

Thanks.

Unfinished sentence: “They are:” was removed.

Comments:

iii. Line 370: “Peak I is assumed reflects”. Reflects what? What is assumed?

Response: It is done.

Thanks.

Peak I is assumed reflects [27,28].

IP phase being directly related to PSI activity, while JI phase parallels the reduction of PQ pool [27,28].

Comments:

  1. Grammatical and spelling errors: The text is full of grammatical errors that could have been avoided. Again the words mentioned below are few examples and are not the only instances.

Response: It is done.

Thanks.

Grammatical errors were corrected.

Comments:

  1. Line 40: proposed should be propose.

Response: It is done.

Thanks.

We proposed propose that [CuL2]Br2 has two binding sites

Comments:

  1. Line 71: “in this case, a weed”, What are the authors referring to when they write “in this case a weed” immediately followed by a paragraph that deals with photosynthetic apparatus, PSII, PSI, etc.

Response: It is done.

Thanks.

We mean that by suppressing the activity of PA or its most vulnerable part, PSII, it is most effective to block the growth of a weed-plant.

We removed (in this case, a weed)

By suppressing the activity of PA (the only source of substances necessary for the growth and development of a plant), or its most vulnerable part, PSII, it is most effective to block the growth of a plant (in this case, a weed).

By suppressing the activity of PA (the only source of substances necessary for the growth and development of a plant), or its most vulnerable part, PSII, it is most effective to block the growth of a plant.

Comments:

iii. Line 307: decrease is misspelled

Response: It is done.

Thanks.

"decrease" was replaced by "reduction"

Comments:

  1. Inconsistency: The authors need to be consistent with defined abbreviations. For example, At many places, the concentration is represented as 3,6 μM (Line 188; including figures), and at other places, they write 3.6 μM (Line 253). Photosystem II is written as PS2 or PSII (Line 67). Photosynthetic apparatus has been defined as PA (Line62-63), but authors spell it out in Line 115.

Response: It is done.

Thanks.

We totally corrected writing of concentrations: everywhere point was replaced by comma.

Line 67

up PA, PS2, and, especially, the oxygen-evolving complex PSII, which oxidizes water, suf

up PA, PSII, and, especially, the oxygen-evolving complex PSII, which oxidizes water,

Line 115

acceptor sides, reaction centre of the PSII as well as about all intermediates of photosynthetic apparatus PA as in intact samples well as under stress impacts, including various inhibitors [27,28].

Comments:

Line 104 describes Fv as variables F(ΔF), what is signified by ΔF or F per se? Section 4.4- DMSO is referred to as DMCO. Sections and sub-sections are numbered incorrectly; Section 2 has sub-sections numbered 4.1 etc. while Section 3 has sub-sections numbered as 2.1 etc. Section 4 has sub-sections numbered 3.1 etc.

Response: It is done.

Thanks.

Line 104

exclusively at the expense of FV (variables F (DF)), which don't recovered by artificial elec

exclusively at the expense of FV, which don't recover by artificial elec

the final concentration of DMCO did not exceed 1%.

the final concentration of DMSO did not exceed 1%.

The numbering of chapters and subsections was totally corrected. Corrected numbering of chapters and subsections is typed by red font and highlighted by green.

Comments:

  1. Factual incorrectness or missing references:
  2. Line 130-131: The authors state 2-3 molecules of PQ-9 per one reaction center and provide references 31 and 32 to back the statement. The references are from the years 1998 and 2000. Since the year 2000, many crystal structures of PSII from various organisms have been resolved, including spinach. To my knowledge only one structure from Thermosynechococcus elongatus (Guskov et al., 2009) shows the presence of a third PQ-9. The third PQ-9 was later proposed as a transient PQ-9 either leaving or entering PSII (Van Eerden et al., 2017). No high-resolution PSII structure since then has identified more than two PQ-9 in PSII. The authors should refer to more recent studies before making such a claim.

Response: It is done.

Thanks.

electron acceptors (PQ-9), approximately 2–3 molecules of PQ-9 per one reaction centre (RC) of PSII [31,32].

electron acceptors (PQ-9), 2 molecules of PQ-9 per one reaction centre (RC) of PSII [31,32].

We changed references 31 and 32. Instead of

  1. Kurreck, J., Renger, G. Investigation of the Plastoquinone Pool Size and Fluorescence Quenching in Photosystem II (PS II) Membrane Fragments. In Photosynthesis: Mechanisms and Effects; Springer Netherlands: Dordrecht, 1998; pp. 1157–1160.
  2. Kurreck, J., Schödel, R., Renger, G. Investigation of the Plastoquinone Pool Size and Fluorescence Quenching in Thylakoid Membranes and Photosystem II (PS II) Membrane Fragments. Photosynth. Res. 2000, 63, 171–182, doi:10.1023/A:1006303510458.

Now

  1. Umena, Y.; Kawakami, K.; Shen, J.-R.; Kamiya, N. Crystal Structure of Oxygen-Evolving Photosystem II at a Resolution of 1.9 Å. Nature 2011, 473, 55–60, doi:10.1038/nature09913.
  2. Suga, M.; Akita, F.; Hirata, K.; Ueno, G.; Murakami, H.; Nakajima, Y.; Shimizu, T.; Yamashita, K.; Yamamoto, M.; Ago, H.; et al. Native Structure of Photosystem II at 1.95 Å Resolution Viewed by Femtosecond X-Ray Pulses. Nature 2015, 517, 99–103, doi:10.1038/nature13991.

Comments:

  1. Section 4.1, Line 139-140: I don’t know what is PSII-membrane. I would understand writing PSII-containing membrane or thylakoid membrane. What is greenhouse spinach?

Response: It is done.

Thanks.

We changed “PSII-membrane” on “PSII-containing membrane” totally everywhere in manuscript.

Specific comments

Comments:

  1. Figure 1 is an exact copy of Figure 1 from the previously published work of the authors (DOI: 10.3390/cells11172680). Moreover, the authors have failed to refer to Figure 1 in the text.

Response: It is done.

Thanks.

We believe that the Figure 1 will help readers and eliminate questions about what is [CuL2]Br2. Therefore, we added next sentence: “To relieve the reader of the need to refer to our previous publication, we present the structure of the ligand (A) and the Cu(II)-complex (B) here again in Figure 1.”

Comments:

  1. Methods: Were the experiments carried out in one day, or the thylakoid membranes were repeatedly frozen and thawed? As the authors may appreciate that multiple freeze-thaw cycles may compromise the integrity of protein complexes.

Response: It is done.

Thanks.

The experiments carried out in one day.

We sincerely apologize for inadvertently misleading you. Sentence: “To explore this mechanism, we investigated the effect of [CuL2]Br2 in the presence/absence of the well-studied inhibitor DCMU on PSII-containing thylakoid membranes by OJIP fluorescence transient measurements” (In Abstract) contains unfortunate misspelling.

It must be as: “To explore this mechanism, we investigated the effect of [CuL2]Br2 in the presence/absence of the well-studied inhibitor DCMU on PSII-containing membranes by OJIP fluorescence transient measurements”

All results (presented in manuscript) were obtained on PSII-containing membranes like BBY-particle not on thylakoids.

Furthermore, in legends to each figure as well as everywhere in text we indicate PSII-containing membranes not thylakoids.

We sincerely apologize for inadvertently misleading you.

Comments:

  1. Section 4.1 does not describe the instrument or the temperature or the light source, and the intensity or optical filter used for the oxygen evolution assay. This information can be critical in determining the overall outcome of the oxygen evolution assay. After reading the section, I was expecting to see the oxygen evolution data but realized that this was done to check PSII integrity in the samples. Authors may want to remove the parts describing oxygen evolution and refer to their previous work to avoid confusion.

Response: It is done.

Thanks.

In the subsection “2.1. Isolation of PSII-containing membranes” we do not describe oxygen evolution assay as method. We only shortly describe method of isolation and properties of PSII-containing membranes. We wanted to show that used bio-samples are nor any special PSII-containing membranes but are routine and normally active PSII-containing membranes.

Comments:

  1. Section 4.2 lacks a description of the dilution methods or buffer or incubation times with inhibitors. Looking at Figure 2, I understand that the buffer possibly contained DCBQ and potassium hexaferricyanide (I might be wrong in assuming so). A clear description of the PAM fluorimeter protocol could help the readers.

Response: It is done.

Thanks.

PAM fluorimeter experiments were carried out in medium containing: in medium containing: 50 mM MES–NaOH (pH 6.5), 300 mM sucrose, 15 mM NaCl. We added insert (in medium containing: 50 mM MES–NaOH (pH 6.5), 300 mM sucrose, 15 mM NaCl,) in next the sentence of text.

The measurements were carried out as follows. A volume of the initial solution of PSII-containing membranes was prepared in medium containing: 50 mM MES–NaOH (pH 6,5), 300 mM sucrose, 15 mM NaCl, and then either an inhibitor solution or the same volume of solvent (in which this inhibitory agent was prepared) was added to an aliquot taken from this volume.

We prepared a stock solutions of inhibitory agents in DMSO taking into account requirement that final concentration of solvent (DMSO) in measuring medium did not exceed 1. Final concentration of the inhibitory agents is indicated in legends.

Fast induction kinetics of chlorophyll fluorescence associated with photoreduction of the PSII primary electron acceptor, plastoquinone QA, were recorded using a MUL-TI-COLOR-PAM fluorimeter (Heinz Walz GmbH, Germany) in a quartz cuvette (optical path length, 1 cm), at room temperature and constant stirring, after adaptation in the dark for at least 15 minutes.

The medium contained no DCBQ and potassium hexaferricyanide.

We believe You think that medium contained DCBQ and potassium hexaferricyanide due to our misspelling in Abstract.

We sincerely apologize for inadvertently misleading you. Sentence: “To explore this mechanism, we investigated the effect of [CuL2]Br2 in the presence/absence of the well-studied inhibitor DCMU on PSII-containing thylakoid membranes by OJIP fluorescence transient measurements” (In Abstract) contains unfortunate misspelling.

It must be as: “To explore this mechanism, we investigated the effect of [CuL2]Br2 in the presence/absence of the well-studied inhibitor DCMU on PSII-containing membranes by OJIP fluorescence transient measurements”

All results (presented in manuscript) were obtained on PSII-containing membranes like BBY-particle not on thylakoids.

Furthermore, in legends to each figure as well as everywhere in text we indicate PSII-containing membranes not thylakoids.

It is known, and we discuss this in the text of the manuscript (with references) lines 234-240, that the OJIP kinetics of PSII-containing membranes do not contain an I peak. Text from manuscript (line 234-240) is below:

  1. Kinetic measured in the absence of additions (control) are completely identical to those recorded on PSII-containing membranes [42–44,49]. There is no peak I in the kinetic (plateau J-I), the main feature characterizing the kinetics of fast chlorophyll fluorescence induction measured on PSII-containing membranes [42–44,49] and therefore the kinetics will be designated below as OJP kinetics [42]. The absence of peak I (plateau J-I) in the OJP kinetics of PSII-containing membranes has been substantiated previously [42].

We sincerely apologize for inadvertently misleading you in Abstract.

We present an additional figure showing the OJIP kinetics measured on thylakoids and on PSII-containing membranes like BBY-particles.

Comments:

  1. A major problem I have with the results is the lack of repeats. The authors should mention if the repeats were performed or not.

Response: It is done.

Thanks.

Each kinetics are average of 5 independent experiments.

We added the sentence in text.

Comments:

The data presented in Figure 3 is probably normalized as (Ft-F0) and not (Ft/F0) as mentioned in Line 222. If my math is correct normalizing by using Ft/F0 should result in a graph that starts at 1 and not 0 on the Y-axis.

Response: It is done.

Thanks.

It is our misprint. It was corrected.

The original OJIP-kinetics normalized relative to F0 are presented as Ft/F0 Ft - F0 versus

Comments:

In lines describing V0I (Lines 379 and 380) F0.02MC and F0.02ms is confusing, again the authors need to be consistent.

Response: It is done.

Thanks.

It is our misprint. It was corrected.

V0I = (Ft – F0.02мс)/(F30мс - F0.02мс)

V0I = (Ft – F0.02ms)/(F30ms - F0.02ms)

Comments:

Figure 5B legend says data is presented in the time range of 0.02-3 ms, but the figure shows up to 30 ms.

Response: It is done.

Thanks.

The time range was corrected.

The difference OJP-kinetics (WOI = VOI exp – VOI cont) plotted in the 0.02-3 ms

The difference OJP-kinetics (WOI = VOI exp – VOI cont) plotted in the 0.02-30 ms

Comments:

Comparing the data presented in Figure 5B and Table 2, the J-step is fixed at 2 ms; however, examining Figure 5B shows that the J peak is shifted beyond 2.5 ms for lines 3 and 4, making the percent changes presented in Table 2 incorrect.

Response: It is done.

It is known that time-positions of peaks in OJIP kinetics may be shifted in samples as a result of impacts. Nevertheless, peaks are still considered to be the same peaks. FJ-peak emits in about 2-3 ms but may shift in longer time. In our case J peak is a little bit shifted beyond 2.5 ms for lines 3 and 4, but we believe that the shift dos not making the percent changes presented in Table 2 incorrect. In Table 2 we estimate value (amplitude) of the J-peak, not its time. The J-peak time-shift beyond 2.5 ms for lines 3 and 4 indicates (as we believe) that the PSII reaction centers are partly destroyed by [CuL2]Br2 and namely therefore appearance of J-peak (reflecting QA photoreduction) is shifted.

Comments:

  1. The discussion section is redundant, unnecessarily lengthy, and ignores the impact of Cu2+ on photosystem II. How about Cu substituting the active metal ions like Mg and Mn at higher concentrations? All discussion is focused on identifying the site for [CuL2]Br2 with no explanation given for the decrease in FM when the organo-metallic complex is present. I will leave the topic to the imagination of the authors to explain the decline of FM.

Response: It is done.

Thanks.

In a previous publication DOI: 10.3390/cells11172680, we showed that the inhibitory effect of [CuL2]Br2 (manifested, among other things, as a concentration-dependent decrease in the FM value solely due to FV) was not associated with disruption of the donor side of PSII. Artificial electron donors did not restore the FM value suppressed by [CuL2]Br2. The chlorophyll concentration did not decrease after incubation PSII-containing membranes at the presence of [CuL2]Br2. The data likely testifies in favor of that Cu in [CuL2]Br2 does not substitute the active metal ions like Mg and Mn.

Comments:

  1. The Bibliography section needs work. Reference 45 and 46-Why are the authors’ names capitalized? References 51 and 52 have texts that are probably not part of the title.

Response: It is done.

Thanks.

  1. Martinazzo, E.G.; Perbini, A.T.; Bacarin, M.A. <b>The Effect of Inhibitors on Photosynthetic Electron Transport Chain in Canola Leaf Discs. Acta Sci. Biol. Sci. 2015, 37, 159, doi:10.4025/actascibiolsci.v37i2.23263.
  2. Martinazzo, E.G.; Perbini, A.T.; Bacarin, M.A. The Effect of Inhibitors on Photosynthetic Electron Transport Chain in Canola Leaf Discs. Acta Sci. Biol. Sci. 2015, 37, 159, doi:10.4025/actascibiolsci.v37i2.23263.
  3. de Carvalho, A.C.; Salvador, J.P.; de M. Pereira, T.; Ferreira, P.H.A.; Lira, J.C.S.; Veiga, T.A.M. Fluorescence of Chlorophyll ≪I≫A≪/I≫ for Discovering Inhibitors of Photosynthesis in Plant Extracts. Am. J. Plant Sci. 2016, 07, 1545–1554, doi:10.4236/ajps.2016.711146.
  4. de Carvalho, A.C.; Salvador, J.P.; de M. Pereira, T.; Ferreira, P.H.A.; Lira, J.C.S.; Veiga, T.A.M. Fluorescence of Chlorophyll a for Discovering Inhibitors of Photosynthesis in Plant Extracts. Am. J. Plant Sci. 2016, 07, 1545–1554, doi:10.4236/ajps.2016.711146.

Comments:

  1. Funding section says “Figure 6 was obtained under the state contract…” Figure 6 in the text is experimental data. Authors need to correct it.

Response: It is done.

Thanks.

Figure 6 was obtained under the state contract of the Ministry of Science and Higher Education of the Russian Federation (theme o. 122050400128-1).

Graphical determination of the initial slope (M0) of the JIP kinetics presented on figure 6 was done under the state contract of the Ministry of Science and Higher Education of the Russian Federation (theme No. 122050400128-1).

==================================================================

Furthermore

Line 91-92

To relieve the reader of the need to refer to our previous publication, we present the structure of the ligand (A) and the Cu(II)-complex (B) here again in Fig. 1.

Line 651

through whole OJP kinetic (but not F0 level) in the presence of our exogenous agent is the

through whole OJP kinetic (but not F0 level) in the presence of our exogenous agent is the

Figure 6. Initial sections of OJP kinetics normalized relative to F0 (F0.02 ms) and maximum fluorescence FM, Vt=(Ft−F0)/(FM−F0)=f(t), on a linear time scale from 0.02 ms to 300 ms in

Figure 6. Initial sections of OJP kinetics normalized relative to F0 (F0.02 ms) and maximum fluorescence FM, Vt=(Ft−F0)/(FM−F0)=f(t), on a linear time scale from 0.02 ms to 0,3 ms in

acceptor sides, reaction centre of the PSII as well as about all intermediates of photosynthetic apparatus PA as in intact samples well as under stress impacts, including various inhibitors [27,28].

Everywhere in the text in decimal fractions the dot was replaced by a comma

Everywhere in the text kinetics was replaced by kinetic
